# Impact Analysis of Worn Surface Morphology on Adaptive Friction Characteristics of the Slipper Pair in Hydraulic Pump

**DOI:** 10.3390/mi14030682

**Published:** 2023-03-19

**Authors:** Siyuan Liu, Chunsong Yu, Chao Ai, Weizhe Zhang, Ziang Li, Yongqiang Zhang, Wanlu Jiang

**Affiliations:** 1Hebei Provincial Key Laboratory of Heavy Machinery Fluid Power Transmission and Control, School of Mechanical Engineering, Yanshan University, Qinhuangdao 066004, China; liusiyuan@ysu.edu.cn (S.L.); aichao@ysu.edu.cn (C.A.); zwz682@stumail.ysu.edu.cn (W.Z.); liziang@stumail.ysu.edu.cn (Z.L.); zyq@stumail.ysu.edu.cn (Y.Z.);; 2State Key Laboratory of Fluid Power and Mechatronic Systems, Zhejiang Univershity, Hangzhou 310027, China; 3Key Laboratory of Advanced Forging & Stamping Technology and Science, Ministry of Education of China Yanshan University, Qinhuangdao 066004, China

**Keywords:** hydraulic pump, slipper pair, adaptive friction, worn surface morphology, fractal theory

## Abstract

The hydrostatic bearing slipper pair of the hydraulic pump has a unique adaptive friction characteristic, which has a better friction reduction and anti-wear ability than the general sliding friction pair, and also has a certain recovery effect on the performance degradation caused by the early wear of the slipper. This paper attempts to reveal the friction adaptive mechanism. Based on the fractal theory, two fractal parameters of fractal dimension and scale coefficient are used to characterize the surface morphology of the slipper mathematically, and the adaptive friction mechanism model is established by combining the friction coefficient equation. The effects of different fractal parameters on the friction coefficient and other performance parameters of slipper pairs are obtained by means of the numerical analysis method. The wear test was carried out by replacing specimens at different intervals to observe the worn surface morphology and the degradation process of the slipper to verify the correctness of the theoretical results. The results show that the friction performance and load-bearing capabilities of the slipper can be recovered to a certain extent within a short period when early wear occurs, and its surface performance shows the variation characteristics of deterioration-repair-re-deterioration-re-repair.

## 1. Introduction

The hydraulic pump is the core power component of the hydraulic system in the fields of national defense, aerospace, and heavy equipment, and the hydrostatic bearing slipper pair is the most critical component and weak link [1,2,3]. A slipper pair is a pair of planes composed of slipper and swash plate, and its motion mechanism and force are very complex. Under the condition of high-speed and heavy-duty operation of the hydraulic pump, it is often faced with severe conditions such as frequent strong load impact, high contact pressure, and fast start, which lead to inevitable wear failure on the surface of the slipper, which decreases the friction performance of the slipper, seriously affects the reliable bearing capacity and working efficiency of the slipper pair, and threatens the stability and service life of the whole pump [4,5]. Different from ordinary sliding friction pairs, the bearing adaptability of slipper pairs in the working process often makes the slipper pairs in the dynamic and static pressure mixed lubrication state, showing complex friction-adaptive characteristics in the early wear stage, which makes the slipper pairs have better anti-wear performance than ordinary sliding friction pairs, and can play a certain role in repairing the early surface wear of slipper pairs [6]. The establishment of a friction model of the mixed lubrication state of slipper pairs is the basis for revealing the friction adaptive mechanism [7,8]. However, at present, the friction adaptive model of slipper pairs has not been established, which causes great difficulties in terms of revealing the failure mechanism of slipper pairs and the reliable bearing design of slipper pairs. Therefore, it is urgent to clarify the friction-adaptive characteristics of the slipper pair in the early wear stage to ensure the safe and stable operation of the hydraulic pump.

Many scholars have carried out a lot of research work on the wear of slipper pair. Rizzo et al. [9] studied the thermal characteristics and friction properties of the slipper pair under oil film lubrication conditions. Gaston et al. [10,11] predicted the damage and fatigue of the slipper pair interface by analyzing the solid deformation, strain, and wear of the slipper and swash plate materials, and tested them through experiments. To improve the wear resistance of the slipper pair, Sohil et al. [12] established a thermal-structural coupling model of the slipper pair under lubrication conditions, and analyzed the heat generation mechanism of the slipper pair. Canbulut et al. [13] studied the influence of friction power loss on the performance of the slipper pair through experiments, and found that reducing friction power loss can improve the performance of the slipper and the efficiency of the axial piston pump. Ibatan et al. [14] combined numerical simulation with experimental research, studied the influence of the surface texture of the slipper on friction performance and bearing performance under different lubrication conditions, and concluded that coating film on the surface of the slipper texture could effectively reduce friction and wear. Ashkan et al. [15] proposed a new design method for the slipper, which can maintain efficiency while reducing wear by optimizing the surface of the slipper. Kou et al. [16] studied the tribological law of dry friction of the slipper pair in an axial piston pump at high temperatures under different loads. Wu et al. [17] studied the structural parameters affecting the friction performance of the slipper, and found that the tribological characteristics of the slipper pair can be optimized by controlling the size, shape, and distribution of pits on the surface of the slipper. Zhang et al. [18] analyzed the surface texture parameters of the hydraulic motor slipper pair and found that the optimized surface texture parameters can reduce the friction coefficient and wear. The adaptive friction property of the hydrostatic support slipper system is reflected in the fact that when the slipper performance is reduced due to wear, it can recover certain performance through its adjustment and adaptation, and in this process, the tribological characteristics of the slipper will change accordingly. There are few studies on the adaptive friction characteristics of the wear process, and the wear mechanism is still unclear. 

The fractal theory is one of the most commonly used methods to study the friction and wear characteristics of friction pairs, especially in ordinary sliding friction pairs. For example, Horovistiz et al. [19] studied the variation in fractal parameters and tribological parameters with surface morphology through friction tests. Wang et al. [20] established a three-dimensional sliding friction pair contact model based on fractal theory and studied the effects of load, rotational speed, and roughness on tribological properties. Dorian et al. [21] studied the interaction between friction surface structure and fractal parameters, and found that there is more excellent wear on the surface of the higher fractals. Agile et al. [22] constructed a fractal prediction model of the static friction coefficient of flange joint surface based on fractal theory, and verified the accuracy of the prediction model by friction test. However, there are few reports on using fractal theory to explore the law of adaptive friction characteristics of the slipper pair.

Therefore, in this study, based on the fractal theory, the fractal dimension and scale coefficient are first used to characterize the surface morphology of the slipper, and the adaptive friction mechanism model is established by combining the oil film stiffness equation, friction coefficient equation, and residual pressing force equation. Then, the model is solved by numerical analysis, and the variation law of friction coefficient, oil film stiffness, and bearing coefficient under different wear degrees are analyzed. Finally, the experimental method of changing specimens in stages is used to observe the variation in wear surface morphology, and the variation law of fractal dimension and scale coefficient in the wear process of the slipper pair is obtained, which verifies the correctness of the model and theoretical analysis results. The influence of fractal parameters on the wear failure process of the slipper pair is analyzed, and the wear mechanism of the slipper pair is revealed.

## 2. Models and Test Methods

### 2.1. Adaptive Friction Characteristic

The slipper pair is a pair of key friction pairs in the hydraulic pump, as shown in Figure 1. The slipper system of hydrostatic bearing structure is mainly composed of a slipper, swashplate, sealing band, central oil chamber, and fixed damping orifice, as shown in Figure 2. The oil flows into the sealing band from the fixed damping orifice through the central oil chamber, where *p*_d_ is the oil supply pressure, *p*_0_ is the pressure of the central oil chamber, *Q* is the leakage flow, *h* is the oil film thickness, *R*_1_ is the inner edge radius of the sealing bar, *R*_2_ is the outer edge radius, *F*_N_ is the pressing force, *F* is the axial thrust, *F*_f_ is the plunger friction force, *F*_a_ is the inertial force, *F*_T_ is the radial force, and *F*_0_ is the supporting force. The slipper designed by the hydrostatic bearing principle is an ideal lubrication condition. In most cases, the design method of residual pressing force is adopted, that is, the design of an incomplete balanced slipper. In this case of *F*_0_ < *F*_N_, the majority of the pressing force *F*_N_ is balanced out by *F*_0_, but there is still a tiny part of the residual pressing force Δ*F* = (*F*_N_ − *F*_0_) to press the slipper on the swashplate, so that the slipper slides close to the surface of the swashplate. This state of residual pressing force is called an underbalanced state. In this state, there is still an oil film on the surface of the slipper pair, but its thickness is thin, and the surface will still have slight friction and wear. If the appropriate residual pressing force can be guaranteed, the slipper can maintain a high total efficiency and a long working life [23].

As can be seen from the figure above, *F*_N_ is the resultant force of axial thrust *F* along the plunger direction and radial force *F*_T_ perpendicular to the plunger direction. The calculation formula of axial thrust *F* acting on the bottom of the plunger by pressure oil is:(1)$$F=\frac{\pi }{4}d_{z}{}^{2}p_{d}$$
where *d_z_* is the plunger diameter.

The relationship between axial thrust *F*, swash plate reaction *F*_N_, radial force *F*_T_ and swash plate inclination *θ* of the plunger pump is as follows:(2)$$F=F_{N}\cos\theta$$
(3)$$F_{T}=F_{N}\sin\theta$$
where *θ* is the swash plate dip angle.

The adaptive friction process of the slipper pair is shown in Figure 3.

The load-bearing adaptive function of the hydrostatic bearing slipper pair is responsible for its adaptive friction characteristic. When the wear intensifies, the bearing capacity can adapt to the change in the wear surface morphology through the independent adjustment of the oil film thickness. At the same time, the slipper pair will also run in the new equilibrium position, so that the bearing performance of the slipper can be restored to a certain extent, and the friction performance of the slipper can be improved. The specific adjustment process is as follows:

When the slipper seal belt is worn, the oil film supporting force *F*_0_ decreases, and the slipper pair pressing force *F*_N_ increases, making the oil film thickness h and the leakage amount *Q* decrease, the pressure *p*_0_ of the central oil chamber increases, the oil film supporting force F_0_ increases, and the change in the autonomous adaptive load reaches a new equilibrium state. At the same time, the contact state of the slipper pair friction surface also changes. A series of physical and chemical changes occur during the contact between the two rough surfaces to adapt to the lubrication environment in the new equilibrium state. The surface morphology of the slip-on boots is changed, and the surface structure with low friction coefficient and wear rate is formed, so that the friction surface presents significant friction-adaptive characteristics.

The process of adaptive friction of the slipper pair occurs in the early wear stage, which refers to the transition stage between the late stage of stable wear and the early stage of severe wear. The friction adaptive mechanism model is suitable for this stage. The stage is represented in the bathtub curve of the friction pair [24], as shown in Figure 4.

### 2.2. Model Establishment

Fractal parameters are used to characterize the surface topography of the slipper pair, and the W-M function [25,26] is used to simulate the surface contour curve of the slipper boots. The two-dimensional W-M function expression is
(4)$$Z(x)=L{\left(\frac{G}{L}\right)}^{d-1}\sum_{n=0}^{n_{\max}}\gamma^{-(2-d)n}cos(2\pi \gamma^{n}\frac{x}{L})$$
where *Z*(*x*) is the surface profile height; *x* is the surface profile displacement coordinate; *d* is the fractal dimension; *n* is the frequency index, *n*_max_=int[log(*L*/*λ*_r_)/Log(*γ*)]; *λ*_r_ is the cutoff wavelength; *G* is the scale coefficient; *L* is the sampling length; *γ* is a constant, and usually 1.5; *γ^n^* indicating the spatial frequency of the contour curve. The fractal dimension and scale coefficient also have the following relationship [27].
(5)$$G={\left[\frac{2R_{q}^{2}}{(\ln\gamma )\sum_{n=0}^{n_{\max}}{\left(\gamma^{n}/L\right)}^{\left(2d-4\right)}}\right]}^{1/(2d-2)}$$
where *R_q_* is the root mean square roughness.

The simplified slipper surface contact model is shown in Figure 5.

It can be seen from Figure 5 that under hydrodynamic lubrication, the height outside the asperity oil film is
(6)$$\Delta h=Z_{\max}-h$$
where *Z*_max_ is the maximum value of the two-dimensional profile.

In Figure 5, *h_n_* is the average surface topography height of the slipper and the gap of the swashplate surface, *R_a_* is the average deviation of the contour arithmetic, and the calculation formula [28] is
(7)$$R_{a}=\frac{1}{L}\int_{0}^{L}\left|Z(x)\right|dx$$

The surface gap *h_n_* is
(8)$$h_{n}=h-R_{a}$$

According to the study of elastic-plastic deformation of asperity [29], the top deformation of asperity *δ* can be expressed by fractal dimension and scale coefficient
(9)$$\delta =G^{1-d}a^{\left(2-d\right)/2}$$
where *a* is the contact area of the asperity.

When the slipper pair enters the early wear stage and wears out, the surface asperity of the slipper pair and the lubricating oil film bear the contact pressure together, the top deformation of asperity *δ* and the oil film thickness is shown in Figure 6.

According to example (6) and example (9), when *δ* = Δ*h*, the contact area of the asperity in the lubrication state is
(10)$$a_{R}={\Delta h}^{2/(2-d)}G^{2/(1-d)(2-d)}$$
where *a_R_* is the contact area of a single asperity.

According to the island area distribution theory [30] and combined with the above analysis, the number of asperities *N* with contact area exceeding the slipper surface *a*_R_ is
(11)$$N\left(A>a_{R}\right)={\left(a_{R}/a\right)}^{d/2}$$

Differentiating the example (11), the area distribution of the asperity is obtained as
(12)$$n\left(a\right)=\left|\frac{dN\left(A>a_{R}\right)}{da}\right|=\frac{d}{2}\frac{a_{R}^{d/2}}{a^{\left(d/2+1\right)}}$$

The total actual contact area is
(13)$$A_{R}=\int_{0}^{a_{R}}n(a)ada=\frac{{d(\Delta h)}^{2/(2-d)}}{2-d}G^{\frac{2}{\left(1-d)(2-d\right)}}$$

Without considering the influence of auxiliary support, the actual contact area of the sealing band surface [31,32] can be expressed as
(14)$$A_{R}=\frac{\pi }{2}\left[\frac{R_{2}^{2}-R_{1}^{2}}{ln\left(R_{2}/R_{1}\right)}\right]$$

Substituting example (6) and example (14) into example (13) yields
(15)$$h=Z_{\max}-{\left\{\frac{\pi }{2}\left[\frac{R_{2}^{2}-R_{1}^{2}}{ln\left(R_{2}/R_{1}\right)}\right]\frac{2-d}{dG^{2/\left(2-d\right)(1-d)}}\right\}}^{\left(2-d\right)/2}$$

According to the study [33], the relationship between the friction coefficient and the surface clearance of the friction pair in the mixed lubrication state is
(16)$$\mu =a_{1}+a_{2}e^{-a_{3}h_{n}}$$
where *a*_1_ is the liquid friction influence coefficient, *a*_2_ is the dry friction influence coefficient, and *a*_3_ is the friction coefficient attenuation index. The corresponding data of friction coefficient and oil film thickness are obtained by experiment, and *a*_1_ = 0.0186, *a*_2_ = 0.1875, *a*_3_ = 0.1620 are calculated.

Residual pressing force on hydrostatic bearing slipper [34] is the difference between pressing force and oil film supporting force
(17)$$\Delta F=F_{N}-F_{0}=\frac{\pi d_{z}{}^{2}}{4\cos\theta }-\frac{\pi (R_{2}^{2}-R_{1}^{2})p_{0}}{2\ln(R_{2}/R_{1})}$$

The relationship between residual pressing force and oil film thickness is
(18)$$\Delta F=m\ddot{h}$$
where m is the slipper quality.

The oil film stiffness represents the change in bearing capacity caused by the change in unit oil film thickness [35], and its relationship with oil film thickness is
(19)$$J=\frac{3A_{R}p_{0}K_{J}h^{2}}{{\left(1+K_{J}h^{3}\right)}^{2}}$$
where *K_J_* is the damping parameter of the slipper pair, $K_{J}=\frac{64l}{3r_{0}^{4}\ln(R_{2}/R_{1})}$, *r*_0_ is the orifice radius of fixed damping, and *l* is the orifice length of fixed damping.

The analysis of example (15) shows that when the slipper pair is worn, the two-dimensional fractal dimension decreases. When the scale coefficient increases, the oil film thickness increasing, the pressure in the central oil chamber of the slipper will fall, and the pressure of the central oil chamber will reduce the supporting force. The analysis of example (14) shows that increasing the oil film thickness will reduce the friction coefficient. In summary, after the wear of the slipper pair, the two-dimensional fractal dimension and scale coefficient will change, resulting in a decrease in the central oil chamber pressure, an increase in the oil film thickness, and a better tribological characteristic.

Combining the above equations yields an adaptive friction mechanism model, as shown below
(20)$$\left\{ \begin{array}{l} h=Z_{\max}-{\left\{\frac{\pi }{2}\left[\frac{R_{2}^{2}-R_{1}^{2}}{\ln\left(R_{2}/R_{1}\right)}\right]\frac{2-d}{dG^{2/\left(1-d\right)\left(2-d\right)}}\right\}}^{\left(2-d\right)/2}\\ \mu =0.0186+0.1875e^{-0.1620h_{n}}\\ F_{N}-F_{0}=m\ddot{h}\\ p_{d}-p_{0}=\frac{8\mu lkh^{3}p_{0}}{\pi r_{0}^{4}} \end{array}\right.$$

### 2.3. Experimental Conditions and Methods

The test stand was built to simulate the wear state of the slide shoe and was mounted on a high-speed wear tester. The speed and load settings for the test could be adjusted by the tester, and the trial rig as a whole is shown in Figure 7. The experimental machine consists of an upper test sample, a lower test sample, and an oil box. The upper test sample simulates the swashplate as shown in Figure 8a and the lower test sample simulates the slipper as shown in Figure 8b.

The schematic diagram of the experimental platform is shown in Figure 9. It consists of the slipper pair friction state simulation device, the data acquisition system, and the oil supply system of the stable oil source. During the experiment, the hydraulic cylinder presses the lower test sample against the upper test sample through the residual pressing force, and the upper test sample rotates at high speed under the action of the servo motor to simulate the friction state of the slipper. The hydraulic system is used as a constant pressure oil source to provide continuous pressure oil to the test sample through the fixed damping holes of the lower test piece, forming a central oil chamber between the upper and lower test samples to produce a lubricating oil film, which ultimately simulates the high speed and heavy load lubrication conditions of the slipper pair [36]. A liquid thermometer is installed in the pump station to monitor the oil temperature in real time. The oil temperature of wear test carried out in this study was always kept in the range of 40~65 degrees, eliminating the influence of temperature on the test. The influence of radial force was considered in the design of the specimen, and the sample with inclination angle was simulated by the swash plate during processing, which was marked in Figure 9.

To analyze the influence of fractal dimension and scale coefficient on the tribological properties of the slipper in the early wear phase of the slipper pair, the experimental method of changing the test sample over time was used, with three lower samples arranged for each test, and the experimental procedure was detailed as follows:(1)Stop the experiment after the first experiment has reached the given wear time, replace the test sample and extend the wear time for a second experiment, and so on.(2)The surface morphology of the slipper is collected at the end of each experiment. The fractal parameters are calculated for each lower sample and averaged to obtain the variation pattern of the fractal parameters throughout the early wear stage.(3)During the experiment, the friction coefficient signal is collected in real-time by the test platform’s sensor with a sampling frequency of 10, and the friction coefficient is observed to change.

According to the above analysis and the actual situation of the experiment machine, a time-shift experiment was designed with a load of 1200 N and a speed of 100 r/min. A total of eight sets of time-shift experiments were carried out. As the frictional adaption process of the hydrostatically supported slipper pair occurs mainly in the early wear phase, eight wear time points were selected from late stable wear to early severe wear. The method for choosing the time points is described below.

First, experiments on the complete frictional wear process of the slipper pair up to failure was carried out, and the obtained friction coefficient signal is shown in Figure 10. Analysis of Figure 10 shows that at around 20,000 s, the wear of the slipper pair reaches the transition from the stable wear phase to the severe wear phase, and after 24,000 s, it enters the tough wear phase. A total of eight-time points were selected from the late stages of stable wear to the early stages of severe wear, as shown in Table 1.

## 3. Results

### 3.1. Numerical Simulation

#### 3.1.1. Parameter Settings

The fractal dimension is set as the disturbance input parameter of the model, and the adaptive friction model is solved by numerical simulation to analyze the changes in the performance parameters of the slipper pair when the slipper wears. The structural parameters set in the simulation are the same as the parameters of the slipper pair of the 10MCY14-1B axial piston pump, which ensures the consistency between the quality of the slipper determined in the simulation and the real slipper. The material of slipper is HMn58-3, and its structure size is shown in Table 2, and the values of the parameters in the initial state are shown in Table 3. 

#### 3.1.2. Analysis of Simulation Results

Setting up different levels of the slipper wear faults at locations with five iterations of the adaptive friction model solution for the slipper system. The fractal dimension of the wear surface is reduced from 1.6, and five different fractal dimensions are set in order from lowest to highest wear level on the slipper surface, the variation rule of various performance parameters of slipper pair when surface wear occurs is analyzed. The values of the fractal dimensions and the corresponding scale coefficients are shown in Table 4.

Figure 11 shows the variation in the friction coefficient of the slipper after different degrees of wear, the friction coefficient before wear was 0.0355. Figure 12 shows the trend of the peak and regulated values of the friction coefficient during the friction adaptation process for different levels of wear.

The friction coefficient is influenced by the surface clearance between the slipper and the swashplate. As can be seen from the change in a single curve in Figure 11, the friction coefficient remains stable before wear. After wear occurs at position 5 of the number of iterations, the friction coefficient increases sharply under the influence of the reduced surface clearance, when the tribological characteristics of the slipper pair deteriorate, after which the friction coefficient increases to a stable state after regrinding of the slipper pair, and the tribological characteristics also become better. A comparison of the five curves illustrated in Figure 11 and combined with Figure 12 shows that the more wear there is, the higher the peak friction coefficient performance and the higher the final stable value of the friction coefficient. The results of this analysis show that although the tribological properties of the slipper deteriorate due to wear, a certain degree of frictional performance can be restored through autonomous adjustment, and that the frictional performance becomes better as the level of wear decreases.

The slipper oil film load-carrying coefficient indicates the load-carrying performance of the slipper pair. Figure 13 shows the variation curve of the load-carrying coefficient during the friction adaption process at different levels of wear, the load-carrying coefficient is 0.81 before wear occurs. Figure 14 shows the trend of the load-carrying coefficient at varying levels of wear during the friction adaption process in terms of valley and adjustment values.

As can be seen from Figure 13, after the number of iterations of 5 position wear, the load-carrying coefficient from a stable value decreases abruptly, at this time the load-carrying performance of the slipper pair is reduced, after the frictional adaption of the slipper system, the load-carrying coefficient returns to a stable value. Combined with the comparative analysis of Figure 13 and Figure 14, it can be seen that as the wear of the slipper surface deepens, the valley value of the load-carrying coefficient becomes lower after wear occurs, and the load-carrying performance of the slipper system becomes worse, and the adjustment value of the load-carrying coefficient also becomes lower, indicating that its load-carrying performance also becomes worse.

Figure 15 shows the variation in the oil film stiffness of the slipper pair after different levels of wear, with an oil film stiffness of 17.79 Pa·s before wear occurs. Figure 16 shows the trend of the peak and regulation values of the oil film stiffness during the friction adaption process for different levels of wear.

Analysis of Figure 15 shows that the oil film stiffness remains stable before wear, and after wear occurs at position 5 of the number of iterations, the oil film stiffness increases to a peak and after that decreases to a steady state, and the new steady state is slightly more significant than the initial state. A comparison of the five curves illustrated in Figure 15 and combined with Figure 16 shows that the greater the level of wear, the higher the peak oil film stiffness performance and the higher the final stable value of the oil film stiffness. The analysis shows that surface wear on the slipper leads to an increase in the oil film stiffness, which increases as the wear level increases.

### 3.2. Analysis of Experimental Results

The friction coefficient values at the end of each phased sample change experiment were used as the friction coefficients at each phased time point, to obtain the friction coefficients in the early wear phase, as shown in Figure 17.

Analysis of Figure 17 shows that the friction coefficient first gradually decreases from around 0.0398 to around 0.0389, then increases to about 0.0396, then drops to about 0.0391. Finally, it begins to increase again, showing a cyclical trend of decrease-increase-decrease-increase again.

The calculated variation curves of the fractal dimension and scale coefficient are shown in Figure 18 and Figure 19.

Analysis of Figure 18 shows that the fractal dimension first gradually increases from 1.61 to about 1.63 and then decreases to about 1.57, showing an overall cyclical trend of increase-decrease-increase-decrease again. This indicates that the slipper pair in the early wear phase has the phenomenon of self-adaptation after the wear occurs, which can make the surface performance of the slipper recover to some extent.

Analysis of Figure 19 shows that in the process of frictional adaption of the hydrostatically supported slipper pair system, the scale coefficient first gradually decreases from 3.25 × 10^−9^ to about 2.84 × 10^−9^ and then increases to about 3.65 × 10^−9^, showing an overall cyclic trend of decreasing-increasing-reducing-increasing again. The change in scale coefficient also illustrates the cyclic trend of deterioration-repair-deterioration-repair of the surface morphology of the slipper at this time.

## 4. Discussion

In summary, the simulation analysis shows that when the slipper surface is worn, the frictional and load-bearing properties of the slipper pair will decrease. The slipper pair can recover part of the frictional and load-bearing properties by the self-adjustment of its own friction-adaptive characteristics, but the oil film stiffness will increase after the slipper wear. As the wear degree of the slipper surface increases, the frictional and load-bearing properties of the slipper pair after wear become worse, and the performance of the slipper pair after adjustment becomes worse. The friction adaptive characteristic of the hydrostatic bearing slipper pair makes it have certain anti-wear performance, but the effect is poor, and with the increase in wear degree, the anti-wear performance becomes worse.

Then through the experimental analysis, it is found that in the early wear stage the surface morphology and friction properties of the slipper show the trend of a deterioration-repair-re-deterioration-re-repair cycle, indicating that the slipper with static pressure bearing structure can have a certain repair effect on the early wear by virtue of its friction-adaptive characteristics. In the process of friction self-adaptation of hydrostatic bearing slipper pair, fractal parameters have a direct impact on the frictional performance. When the fractal dimension and characteristic roughness increase, the frictional performance will improve; when the fractal dimension and characteristic roughness decrease, the frictional performance will deteriorate; on the contrary, when the scale coefficient increases, the frictional performance will deteriorate. The frictional properties will be better.

## 5. Conclusions

In this paper, the frictional adaptive mechanism model of the early wear stage of the slipper pair is established; the variation pattern of the frictional adaptive characteristics of the slipper pair during the change in the fractal dimension and scale coefficient is analyzed; the wear mechanism of the slipper shoe pair is revealed. The specific conclusions are as follows:(1)Based on the fractal theory, the mathematical mapping relationship between oil film thickness and surface morphology fractal characterization parameters is derived, and the adaptive friction mechanism model of the slipper pair is established by combining the oil film stiffness equation, the friction coefficient equation, and the residual pressing force equation.(2)The numerical solution of the model reveals that when early wear occurs, the load-carrying coefficient becomes smaller, and the friction coefficient and oil film stiffness become more extensive, resulting in poorer load-carrying performance and tribological properties, which can be restored after adaptive adjustment of the slipper pair, and the load-carrying performance and tribological properties of the slipper pair will gradually become better as the fractal dimension becomes more extensive and the scale coefficient becomes smaller.(3)The correctness of the simulation results has been verified through experiments. The surface morphology and tribological properties of the slipper pair in the early wear phase show a cyclic change process of deterioration-repair-deterioration-repair, and the frictional properties become better when the fractal dimension becomes more extensive, and the scale coefficient decreases.

## Figures and Tables

**Figure 1 micromachines-14-00682-f001:**
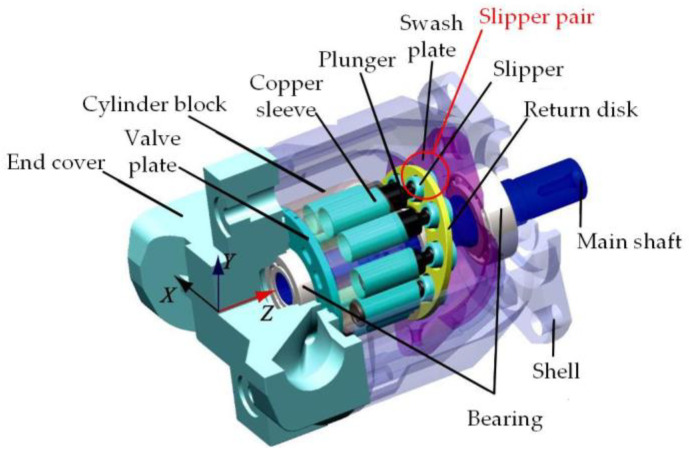
Hydraulic pump structure drawing.

**Figure 2 micromachines-14-00682-f002:**
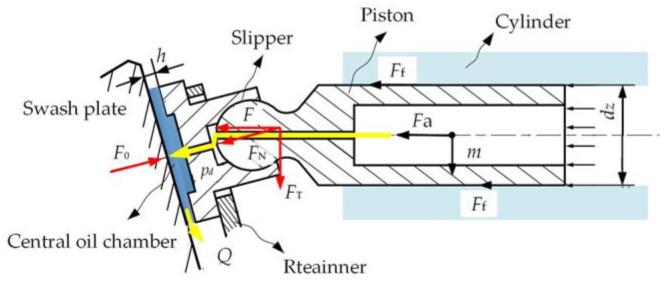
Structure of hydrostatic bearing slipper.

**Figure 3 micromachines-14-00682-f003:**
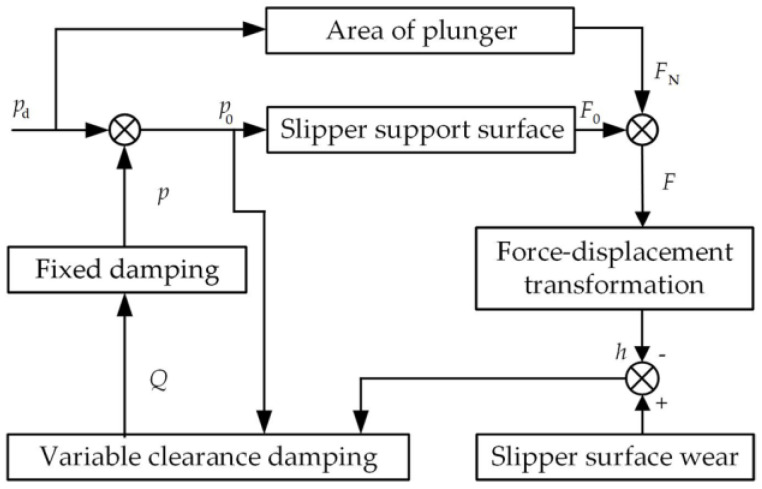
Adaptive friction adjustment of the slipper pair.

**Figure 4 micromachines-14-00682-f004:**
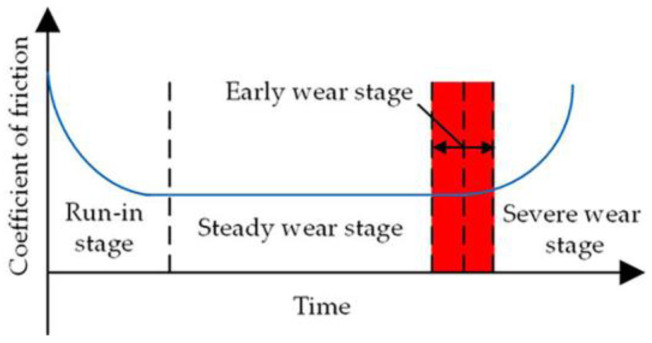
Early wear stage diagram of the slipper pair.

**Figure 5 micromachines-14-00682-f005:**
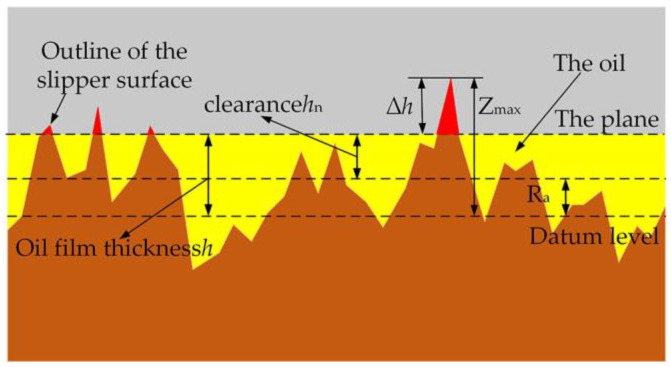
Simplified contact model of the slipper surface.

**Figure 6 micromachines-14-00682-f006:**
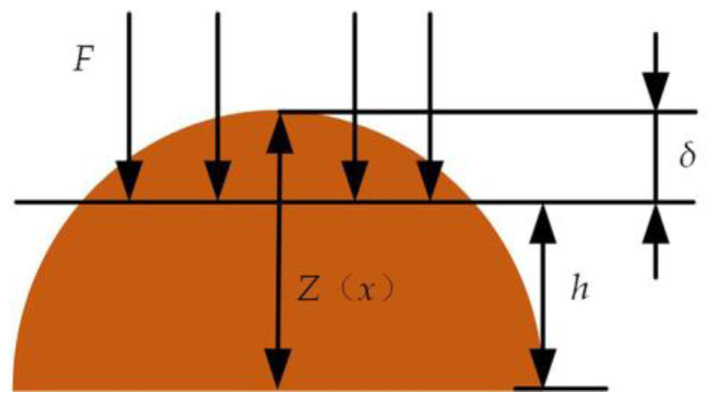
Simplified contact model of asperities on slipper surface.

**Figure 7 micromachines-14-00682-f007:**
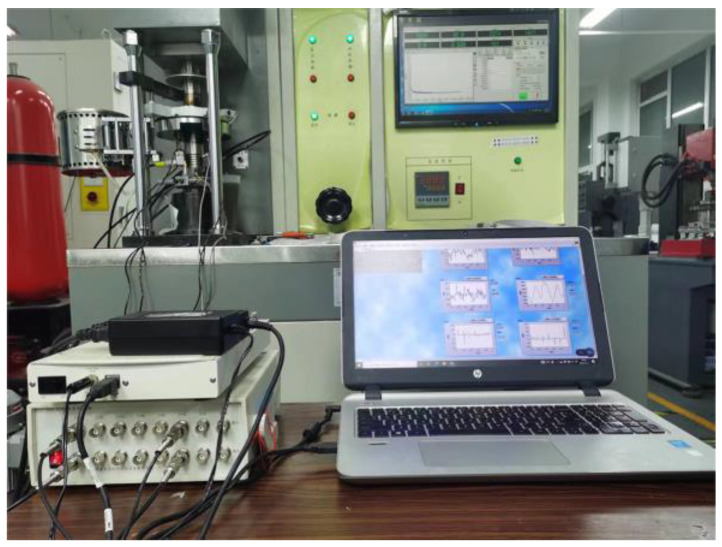
Experimental platform.

**Figure 8 micromachines-14-00682-f008:**
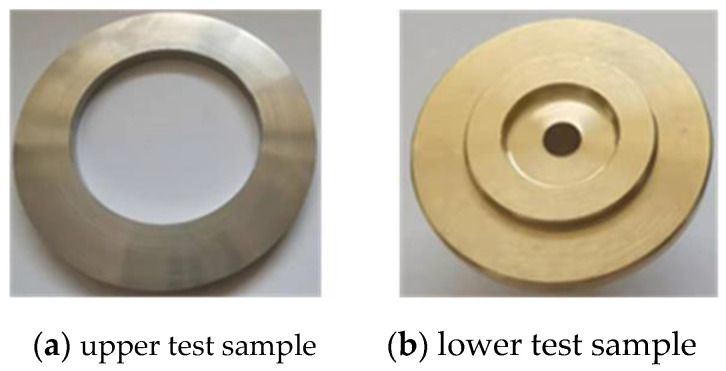
Test sample used for experiments.

**Figure 9 micromachines-14-00682-f009:**
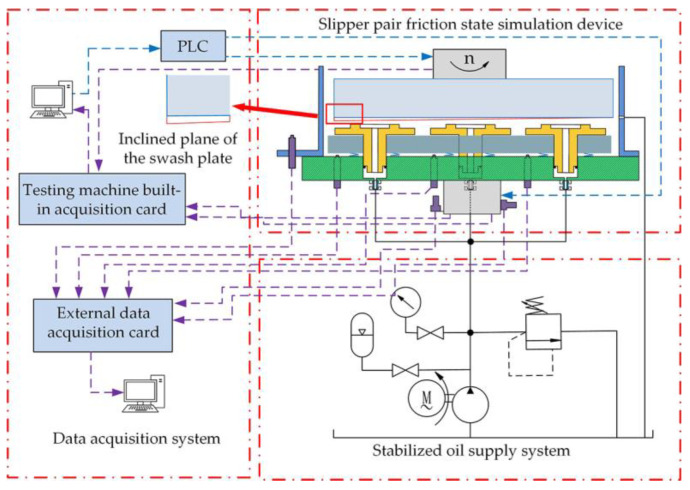
Schematic diagram of the experimental platform.

**Figure 10 micromachines-14-00682-f010:**
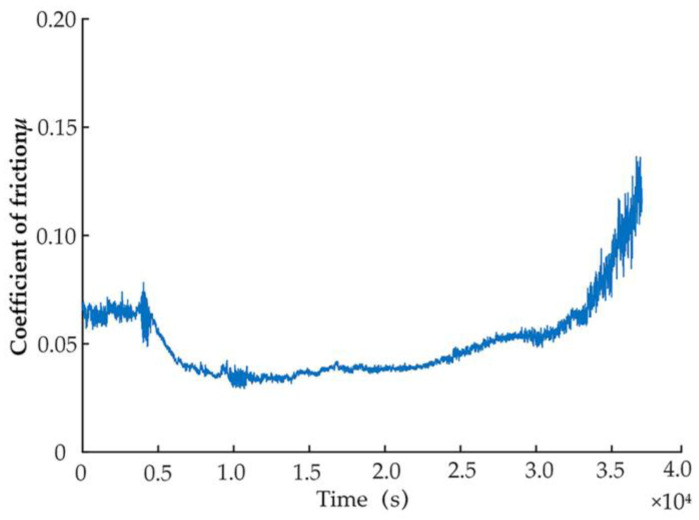
Friction coefficient signal.

**Figure 11 micromachines-14-00682-f011:**
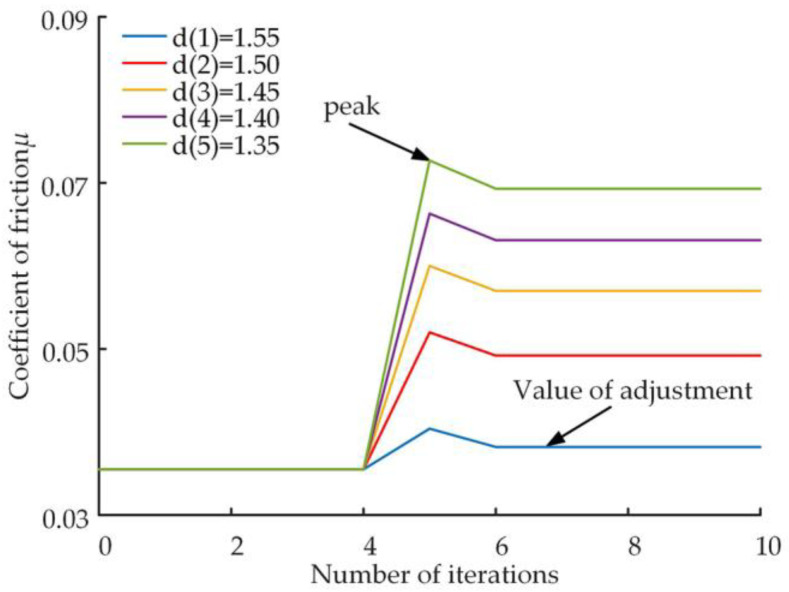
Friction coefficient curve.

**Figure 12 micromachines-14-00682-f012:**
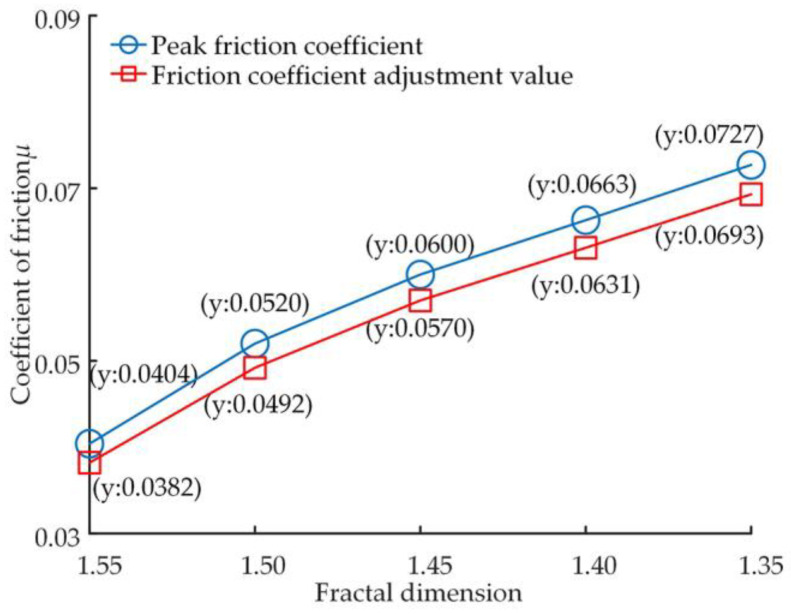
Variation curves of the friction coefficient valley and regulation values.

**Figure 13 micromachines-14-00682-f013:**
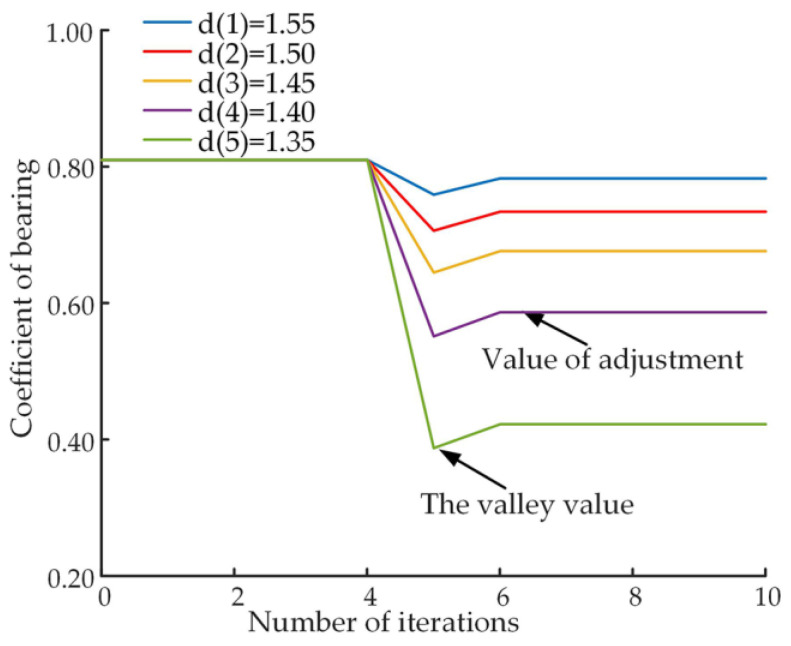
Variation curve of bearing coefficient.

**Figure 14 micromachines-14-00682-f014:**
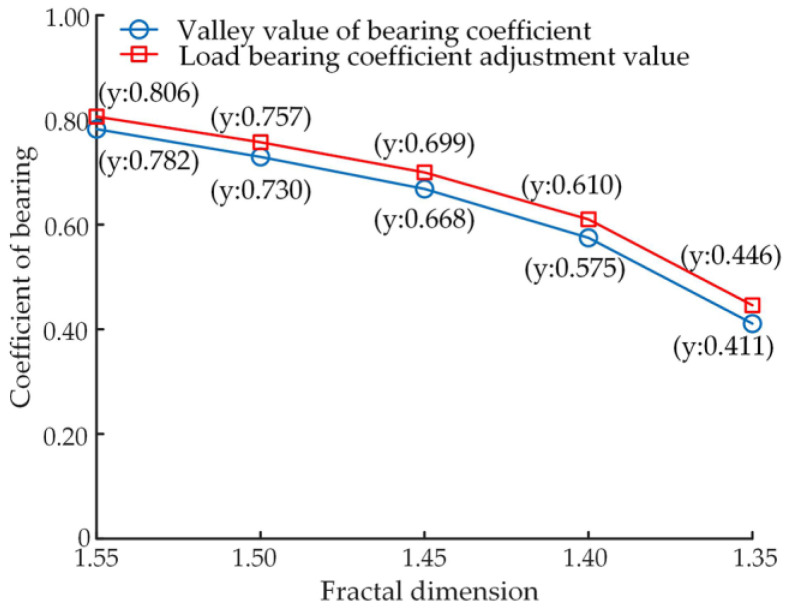
Variation in the valley and regulation values of the load-carrying coefficient.

**Figure 15 micromachines-14-00682-f015:**
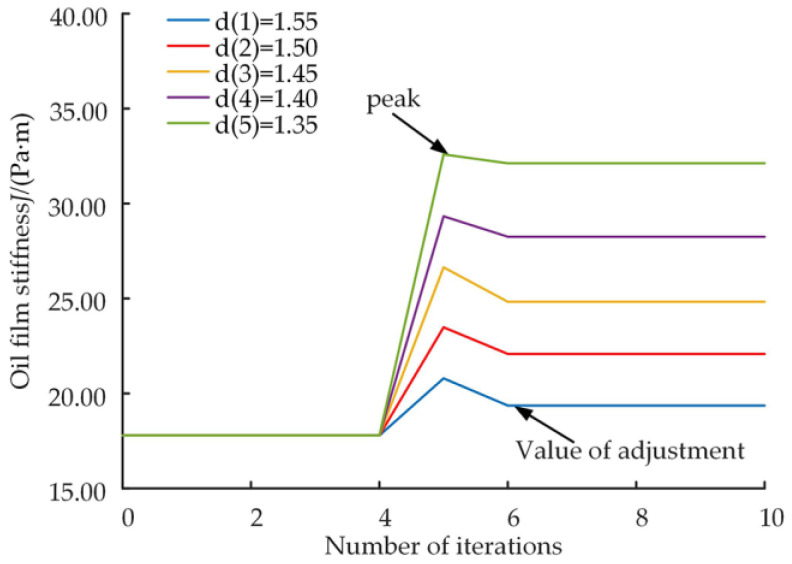
Oil film thickness curve.

**Figure 16 micromachines-14-00682-f016:**
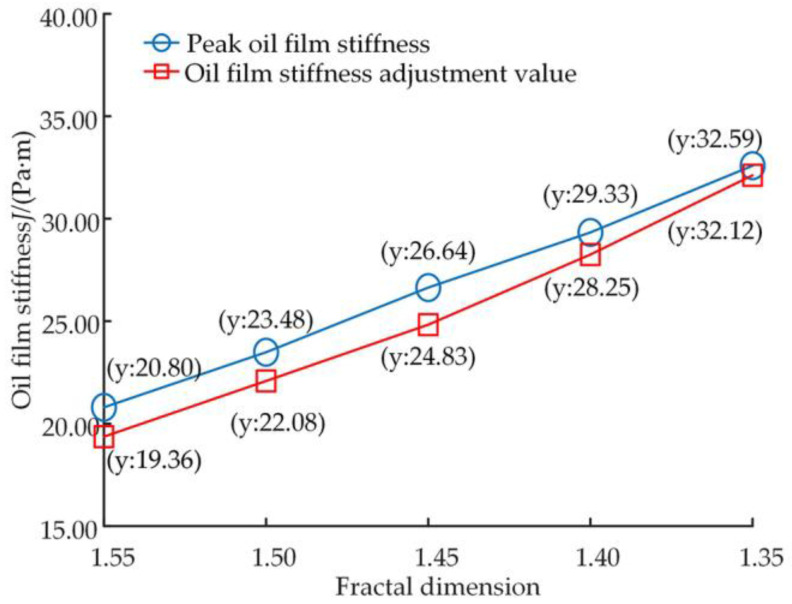
Variation curves of peak oil film stiffness and regulation values.

**Figure 17 micromachines-14-00682-f017:**
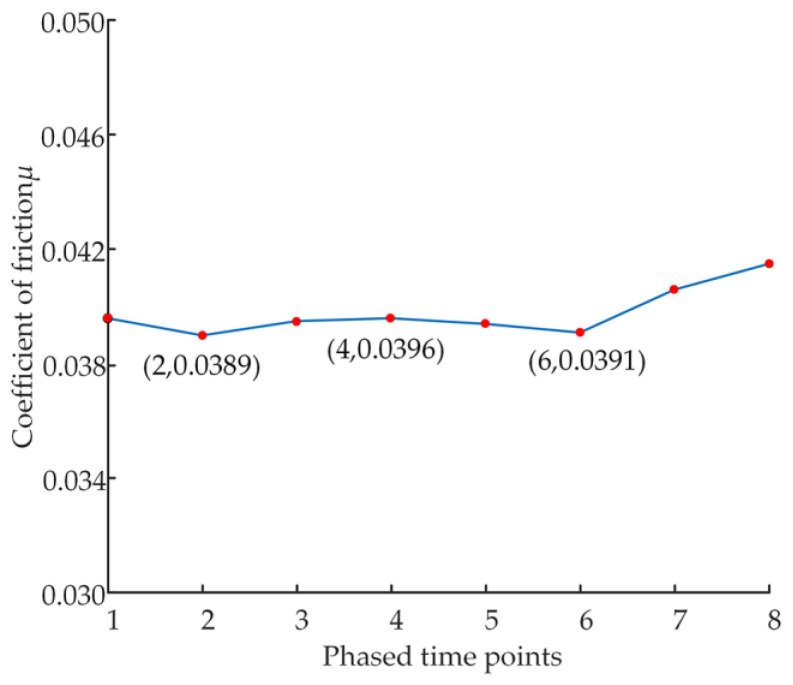
Friction coefficient curve.

**Figure 18 micromachines-14-00682-f018:**
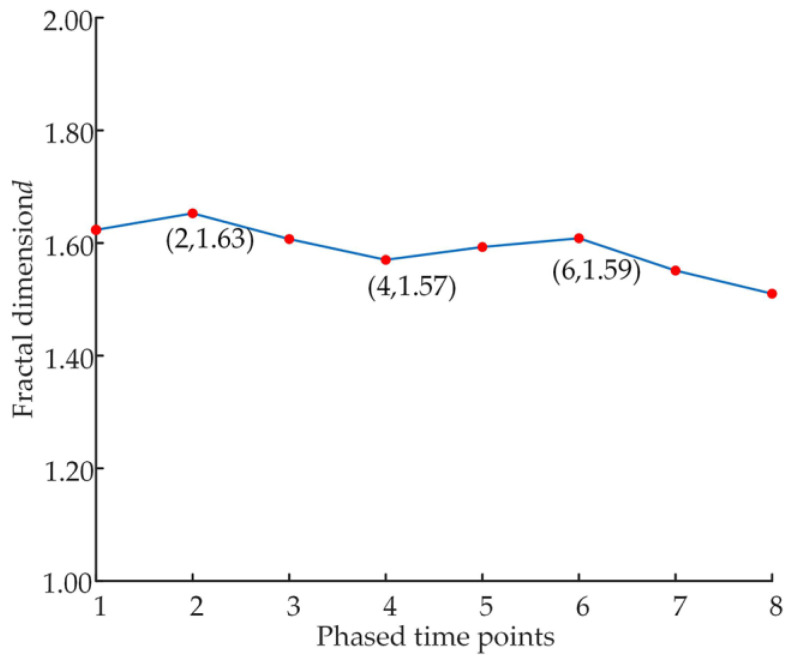
Variation in fractal dimension.

**Figure 19 micromachines-14-00682-f019:**
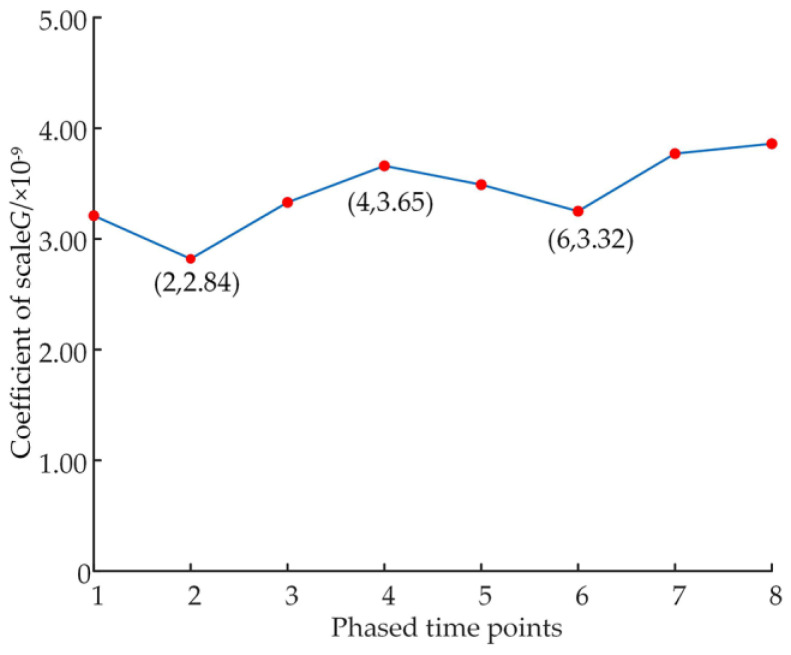
Variation in scale coefficient.

**Table 1 micromachines-14-00682-t001:** Timepoint selection table.

Time Segment Time Point	Time/s	Time Segment Time Point	Time/s
1 2 3 4	22,000 22,500 23,000 23,500	5 6 7 8	24,000 24,500 25,000 25,500

**Table 2 micromachines-14-00682-t002:** Structural and operating parameters of the slipper system.

Parameter of Structure	The Numerical	Parameter of Structure	The Numerical
The slipper quality Inner edge radius Outer edge radius Plunger diameter	50 g 0.009 m 0.016 m 0.023 m	Viscosity of oil Damping hole radius Damping hole length cos *θ*	0.026 Pa·s 0.001 m 0.12 m 0.66

**Table 3 micromachines-14-00682-t003:** Values for the initial state parameters of the slipper system.

Parameter	The Numerical	Parameter	The Numerical
Fractal dimension Scale coefficient	1.60 0.0025	Coefficient of bearing Oil film thickness	0.83 7.23 μm

**Table 4 micromachines-14-00682-t004:** Value of the fractal parameter.

Fractal Parameter	Numerical Size				
Fractal dimension Scale coefficient	1.55 0.0044	1.50 0.0074	1.45 0.0122	1.40 0.0191	1.35 0.0217

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
