# Peer review of "Impact Analysis of Worn Surface Morphology on Adaptive Friction Characteristics of the Slipper Pair in Hydraulic Pump"

_micromachines, 2023, doi:10.3390/mi14030682_

Round 1
Reviewer 1 Report
The author analyzes the influence of worn surface morphology on the adaptive friction characteristics of hydraulic pump slippers. The research results provide some information on the change in the wear process of hydraulic pump slippers with time. The results show that after the early wear of the slippers, their friction performance and bearing capacity have been restored to a certain extent in a short time, and their surface properties show the characteristics of deterioration - repair - re-deterioration - re-repair. However, there are still some problems. The specific problems are as follows:
1. In Figure 1, the hydraulic pump sliding shoe is thrust by the plunger in the axial direction of the plunger, and the sliding shoe is subjected to the component force in the radial direction of the swashplate. The influence of periodic radial force on the sliding shoe is not considered in the paper.
2. In 2.3 Test conditions and methods, the viscosity of hydraulic oil gradually decreases with the increase in temperature. The friction temperature of hydraulic oil increases rapidly in the actual process. Is the temperature controlled during the test, and the influence of temperature considered?
3. Is the Scale factor the same as the Scale coefficient? Please use a unified name.
4. In the analysis of experimental results in 3.2, it was found that in the early stage of wear of the sliding shoe pair, the surface morphology of the sliding shoe showed a cycle trend of deterioration - repair - deterioration - repair, which had a certain repair effect on the early wear. Please analyze the physical reasons for the adaptive phenomenon of early wear.
Author Response
The author analyzes the influence of worn surface morphology on the adaptive friction characteristics of hydraulic pump slippers. The research results provide some information on the change in the wear process of hydraulic pump slippers with time. The results show that after the early wear of the slippers, their friction performance and bearing capacity have been restored to a certain extent in a short time, and their surface properties show the characteristics of deterioration - repair - re-deterioration - re-repair. However, there are still some problems. The specific problems are as follows:
Reply: Thank you for your detailed suggestion. According to the suggestions, the author has checked and modified the English of the manuscript, and improved the results and conclusions. The following is the author's reply to your specific questions.
1、In Figure 1, the hydraulic pump sliding shoe is thrust by the plunger in the axial direction of the plunger, and the sliding shoe is subjected to the component force in the radial direction of the swashplate. The influence of periodic radial force on the sliding shoe is not considered in the paper.
Reply: Thank you for your detailed suggestion. In the revised draft, the author adds the analysis of the radial force of the slipper on the swash plate, please see the attachment for details.
2、 In 2.3 Test conditions and methods, the viscosity of hydraulic oil gradually decreases with the increase in temperature. The friction temperature of hydraulic oil increases rapidly in the actual process. Is the temperature controlled during the test, and the influence of temperature considered?
Reply: Thank you for your detailed suggestion. The test bed designed by the author fully considers the problem of heat dissipation in the test process, and a liquid thermometer is installed in the pump station, which can monitor the oil temperature in real time. In the wear test conducted in this study, the oil temperature was always kept within the range of 40-65 degrees, so the influence of oil temperature on the experimental results was not considered in the test. Please see the attachment for details.
3、 Is the Scale factor the same as the Scale coefficient? Please use a unified name.
Reply: Thank you for your detailed suggestion. In this paper, both scale factor and scale coefficient represent scale coefficient, which has been modified by the author and unified in the manuscript.
4、 In the analysis of experimental results in 3.2, it was found that in the early stage of wear of the sliding shoe pair, the surface morphology of the sliding shoe showed a cycle trend of deterioration - repair - deterioration - repair, which had a certain repair effect on the early wear. Please analyze the physical reasons for the adaptive phenomenon of early wear
Reply: Thank you for your detailed suggestion. The physical reason for this phenomenon is: When the slipper seal belt is worn, the oil film supporting force F0 decreases, and the slipper pair pressing force FN increases, making the oil film thickness h and the leakage amount Q decrease, the pressure P0 of the central oil chamber increases, the oil film supporting force F0 increases, and the change of the autonomous adaptive load reaches a new equilibrium state. At the same time, the contact state of the slipper pair friction surface also changes. A series of physical and chemical changes occur during the contact between the two rough surfaces to adapt to the lubrication environment in the new equilibrium state. The surface morphology of the slip-on boots is changed, and the surface structure with low friction coefficient and wear rate is formed, so that the friction surface presents significant friction adaptive characteristics. Please see the attachment for details.

Reviewer 2 Report
The Introduction section should be improved. In its current form, it does not show the need for the new research presented in the reviewed paper. References should be supplemented. Some figures are redundant - they are obvious (figs. 2, 5). Notes are included in the text of the manuscript.

Author Response
The Introduction section should be improved. In its current form, it does not show the need for the new research presented in the reviewed paper. References should be supplemented. Some figures are redundant - they are obvious (figs. 2, 5). Notes are included in the text of the manuscript.
Response: Thank you for your detailed suggestion. In response to your suggestion, the author has supplemented the research background aspects of the introduction in detail and added references, the serial numbers of the references listed below are consistent with the revised version. which are detailed in the reply to Note one below.
Additional references:
- Gang, L.; Huang, G.; Yuan, H.; Xia, S.; Tan, W. Analysis and optimisation of impact wear of diesel engine needle valve assembly. J. Hydromechatronics 2022, 5(1), 80–91, doi: 10.1504/IJHM.2022.10046787.
- Danes, L.; Vacca, A.;A frequency domain-based study for fluid-borne noise reduction in hydraulic system with simple passive elements. J. Hydromechatronics, 2021, 4(3), 203–229, doi: 10.1504/IJHM.2021.10037156.
- Stosiak, M.; Karpenko, M.; Deptuła, A.; Urbanowicz, K.; Skačkauskas, P.; Cieślicki, R.; Deptuła, A.M. Modelling and Experimental Verification of the Interaction in a Hydraulic Directional Control Valve Spool Pair. J. Applied Sciences-Basel. 2023, 13(1), 458, doi:10.3390/app13010458.
- Taylor, R, I.; Sherrington, I. A Simplified Approach to the Prediction of Mixed and Boundary Friction. J. Tribology International, 2022, 175, 107836, doi: 10.1016/j.triboint.2022.107836.
Figures 2 and 5 are the basis for the establishment of the friction adaptive model at the early wear stage. They are not explained in detail in the original manuscript, but have been supplemented by the author in the revised draft,Please see the attachment for details.
Note 1. The friction model is described in this paper Stosiak, M.; Karpenko, M.; Deptuła, A.; Urbanowicz, K.; Skačkauskas, P.; Cieślicki, R.; Deptuła, A.M. Modelling and Experimental Verification of the Interaction in a Hydraulic Directional Control Valve Spool Pair. Appl. Sci. 2023, 13, 458. https://doi.org/10.3390/app13010458.
Response1: Thank you for your detailed suggestion. According to the suggestions, this reference is added to the manuscript and the research background is modified (See reference [7]). Please see the attachment for details.
Notes 2 and 4. How to determine the quality of slipper used in simulation?
Response2and4: Thank you for your detailed suggestion.
The structural parameters set in the simulation are the same as the parameters of the slipper pair of the 10MCY14-1B axial piston pump, which ensures the consistency between the quality of the slipper determined in the simulation and the real slipper. Please see the attachment for details.
Note 3. KJ Is this parameter determined or just from the literature?
Response3: Thank you for your detailed suggestion. KJ is the damping parameter of the slipper, which is determined by the inner edge radius R1, outer edge radius R2, damping hole radius r0 and length l of the damping hole of the slipper pair seal belt. In this paper, R1, R2, r0 and l are confirmed values, which are 0.009m, 0.016m, 0.001m and 0.12m respectively. See Table 2 of the manuscript for details. KJ is confirmed value in this paper. This formula is derived from the following references, which the author has supplemented in the manuscript.Please see the attachment for details.
- Wang, C. Dynamic characteristics analysis of axial piston pump slipper pair with wear fault. Master's thesis, Yanshan Uni-versity, Qinhuangdao, China, May 2017.

Round 2
Reviewer 1 Report
The author answered my question. I suggest that the paper can be published.